# Biomedical Applications of Blow-Spun Coatings, Mats, and Scaffolds—A Mini-Review

Mohammadmahdi Mobaraki [1], Meichen Liu [2], Abdul-Razak Masoud [3,4] and David K. Mills [3,4,*]

1 Faculty of Biomedical Engineering, Amirkabir University, Tehran 15916-34311, Iran
2 Faculty of Biomedical Engineering, Louisiana Tech University, Ruston, LA 71272, USA
3 Molecular Science and Nanotechnology, Louisiana Tech University, Ruston, LA 71272, USA
4 The School of Biological Sciences, Louisiana Tech University, Ruston, LA 71272, USA
* Correspondence: dkmills@latech.edu; Tel.: +1-318-257-2640; Fax: +1-318-257-4574

**Abstract:** Human tissues and disease models require well-defined biomimetic microenvironments. During the past decade, innovative developments in materials science, microfabrication, and polymer science have provided us with the ability to manipulate cellular microenvironments for regenerative medicine and tissue engineering applications. Solution blow spinning is a facile fiber fabricating method that requires a simple apparatus, a concentrated polymer solution within a volatile solvent, and a high-pressure gas source. Commercially available airbrushes, typically used for painting and cosmetic makeup, have successfully generated a range of nanofibers and films. Applications under investigation are similar to electrospinning and include enzyme immobilization, drug delivery, filtration, infection protection, tissue engineering, and wound healing. This review will discuss fiber fabrication methods before a more detailed discussion of the potential of blow-spinning in biomedical applications.

**Keywords:** blow-spun coatings; mats; scaffolds

## 1. Introduction

Each year, countless individuals experience damage to tissues and organs due to congenital disabilities, trauma due to accident, violence, war, or disease; thus, significant clinical demand exists to promote the regeneration of injured/diseased tissues [1,2]. Furthermore, innovative developments in 3D printing, materials science, polymer chemistry, biofabrication, and associated technologies have provided us with the tools to better mimic the in vivo cellular microenvironment [3,4]. As a result, the potential for biomaterial-based therapeutic approaches in regenerative medicine is being realized, with significant research emphasis on developing local drug delivery systems (DDS), customized treatments, and biomimetic methods for tissue regeneration [5,6].

The extracellular matrix (ECM) strongly influences cellular responses and regulates cell adhesion, proliferation, migration, and differentiation [7–9]. A bioengineered tissue scaffold that mimics the natural ECM will create a microenvironment that significantly enhances cellular response [10,11]. This recognition has increased the study of decellularized ECMs used directly as a bioscaffold for drug delivery or tissue engineering [10–12]. These native ECM scaffolds provide structural support and possess instructive bioactive signals that direct cellular behavior [13–15]. Native ECM has also been derived from cultured cells and used as tissue scaffolds, but these often lack the proper form and structural support [16]. However, these ECM scaffolds suffer from several drawbacks, including differences in size and conformation compared to the original tissue, lack of micro-tailored geometry, tuneability, the inclusion of instructional agents, and the potential risk of introducing pathogens.

The ideal replacement scaffold should mimic the structural and functional profile of the materials found in the native ECM of the target tissue or organ [17]. The scaffolding must also support and possess a three-dimensional organization, optimal porosity, induce cellular

differentiation, enable phenotypic maintenance for cells seeded within the scaffold, and create a microenvironment that maintains cell function within the cellular compartment [18]. In the following sections, we discuss the current, state-of-the-art research, utility of blow spinning in biomedicine with a focus on applications in bone and skin tissue regeneration, medical textiles, and wound healing.

## 2. Solution Blow Spinning

Electrospinning and electrospraying are commonly used methods for creating nanofiber mats and scaffolds [19–22]. Electrospinning is easily adaptable for the creation of most tissue architectures and offers the ability to use many polymer types and polymer composites. It is also scalable, enables defined fiber organization, porosity and well as tunablity in terms of the inclusion of bioactive factors [22,23]. Nanofiber coatings, mats, and scaffolds have many biomedical applications, including drug delivery [23], wound dressings [24,25], tissue engineering [26,27], and medical textiles [28–30].

Electrospinning is a powerful and widely studied technique. It requires specialized equipment, high voltages, and electrically conductive targets [31]. Furthermore, it also suffers from a relatively low deposition rate, use of toxic solvents, and the end product depends on many variables. These restrictions prohibit the use of electrospinning for any in situ deposition of fibers in surgery or for conformal coverage of nonconductive targets without the use of polymer melts or the assistance of air flow [31,32].

Solution blow spinning offers an easily adaptable alternative that can generate on-demand nanofiber mats directly on a wide range of targets [30,31]. Previous studies have described its ease of use and rapid deposition rate as compared to electrospinning [32–34]. This technique has been studied for various applications, including drug delivery, microfiltration, and tissue engineering.

Solution blow spinning uses a commercially available airbrush (or equivalent) and compressed gas to generate nanofilms or nanofibers or nanofiber/film composites (Figure 1). Generating fibers and films through solution blow spinning is a high-throughput alternative technique requiring a straightforward set-up: a concentrated polymer solution in a volatile solvent and an airbrush fitted to a high-pressure gas source. The solution blow-spin method allows for the direct deposition of nanofiber meshes and scaffolds onto various geometries [32,33]. The concept provides for on-demand fabrication of conformal nanofiber mats and allows for precise and site-specific construction. The approach may be used as a surgical sealant in place of or in addition to sutures in vascular, intestinal, or airway anastomosis applications [32]. Solution blow spinning could also be useful in areas requiring the use of a hemostatic material or sealant, especially when large areas are exposed, and conventional suturing may not be possible [33]. Airbrushing is a simple way to deposit large areas of catalyst particles. Airbrushing also produces a relatively uniform coating of the substrate, and it can be applied to a large extent at room temperature in a short period, with various polymers [34,35].

The solution blow spinning (SBS) technique can produce micro- and nanofibers of a wide variety of polymers at a higher fiber output than traditional electrospinning techniques. The SBS spinning system consists of concentric nozzles through which a polymer solution and a pressurized gas are simultaneously ejected. The nozzle tip, the aerodynamic drag, and shear forces caused by the pressurized gas exiting the nozzle are combined to form a cone similar to the Taylor cone formed at the nozzle tip in the electrospinning process. Fiber bundles are jettisoned from the nozzle tip in the SBS process towards a collector placed at a fixed working distance [26,35,36]. SBS has been successfully used in controlled-release studies, including progesterone and linalool [37,38].

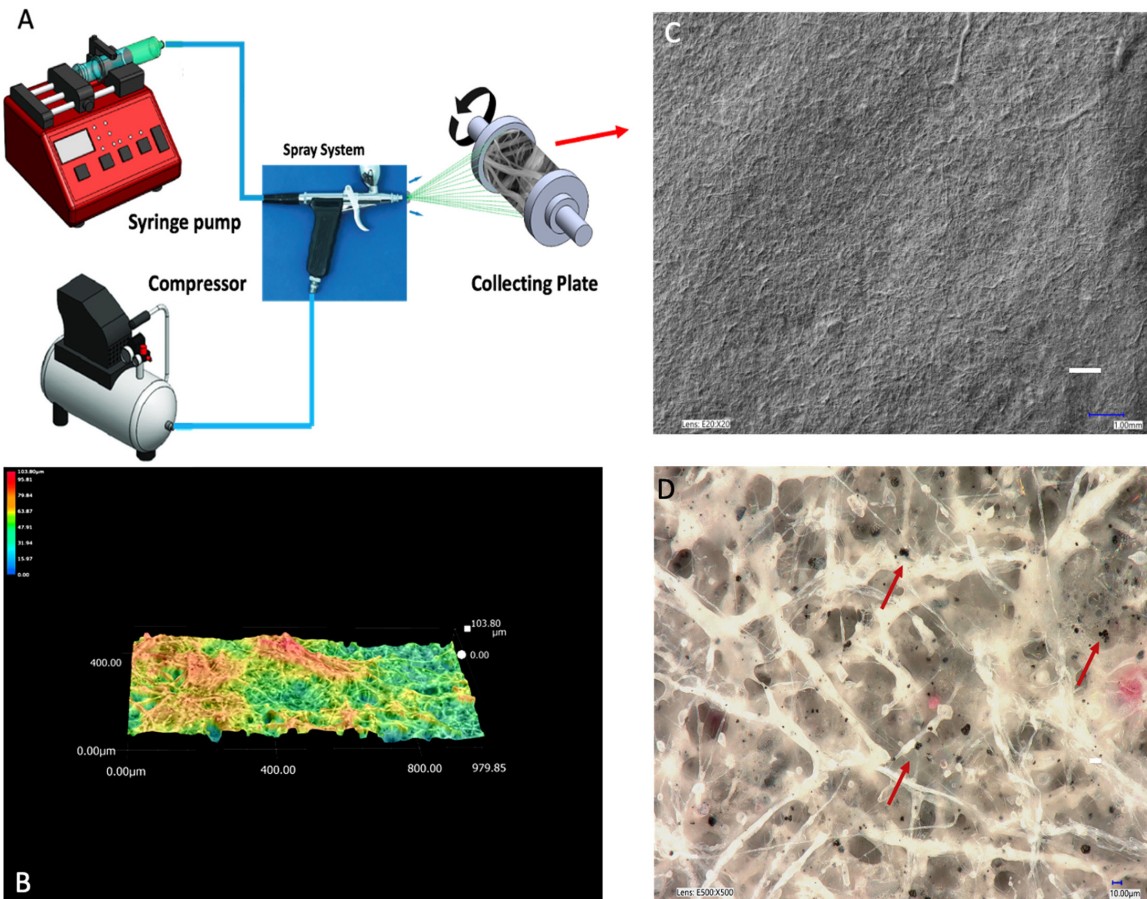

**Figure 1.** (**A**) Typical blow spinning set-up. (**B**) Surface topography map of a blow-spun textile composed of PCL and zinc-coated halloysite nanotubes (ZnHNTs). (**C**) Digital micrograph of the surface of a PCL/ZnHNT blown spun fabric. White bar equals 1 mm. (**D**) High power view of PCL fibers with embedded zinc-coated halloysite nanotubes (back specs) indicated by red arrows). White bar equals 1 μm. All figures are obtained from the corresponding author's collection.

Blow spinning applications include many that often use electrospinning as the fabrication method and include enzyme immobilization [39], drug delivery [40], tissue repair [41], and regenerative medicine [42] (Figure 2). The reported examples of this technique cite the ease of use and rapid deposition rate compared to electrospinning; Tutak et al. were the first to move toward using the method to generate conformal coatings for tissue engineering scaffolds [42]. Behrens et al. (2014) used solution blow spinning to fabricate conformal mats of poly(lactic-co-glycolic acid) (PLGA) in situ using a typical set-up of a commercial airbrush using compressed $CO_2$ [32]. Their study showed the facile fabrication of PLGA nanofibers using only a commercial airbrush and compressed $CO_2$. The solution and deposition conditions were optimized, and mechanical properties were characterized. Nanofiber degradation was monitored over 42 days for molecular weight and morphology changes in vitro. The biocompatibility of direct nanofiber deposition was quantitatively assessed using a cell viability assay, and blood interaction was assessed qualitatively with SEM. A pilot animal study was then used to demonstrate possible surgical applications, including a surgical hemostatic, a surgical sealant, and tissue reconstruction [32].

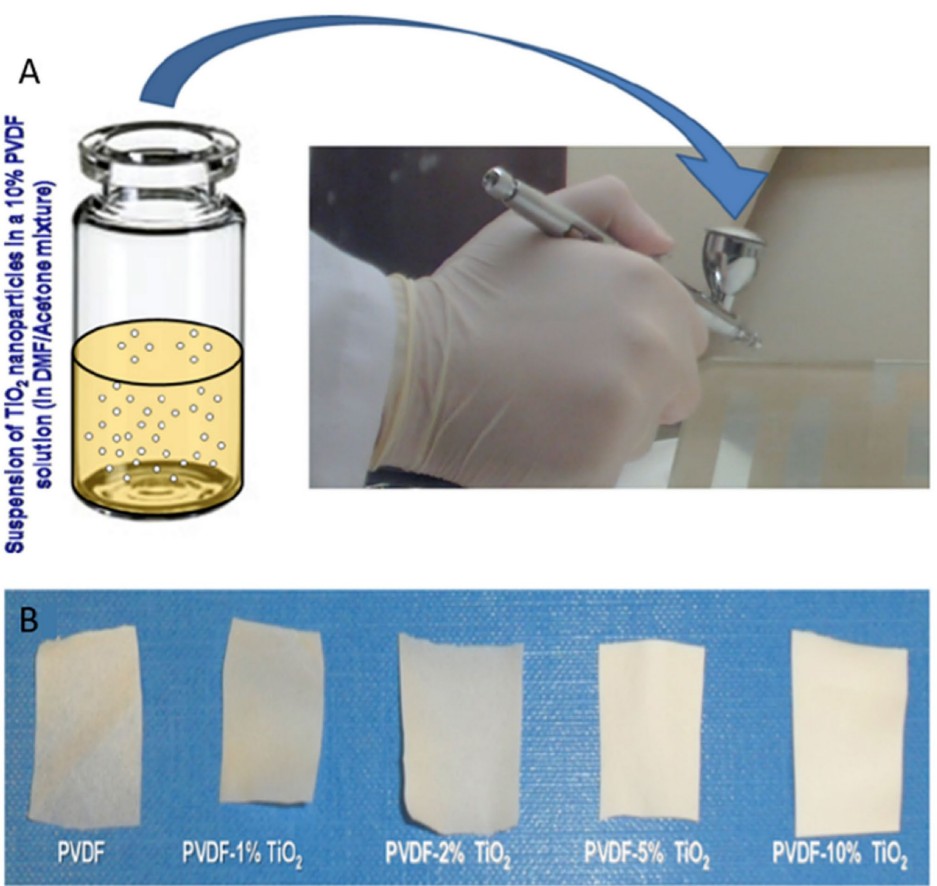

**Figure 2.** (**A,B**) Solution blow spinning was used to prepare nanocomposites based on PVDF filled with well-dispersed titanium dioxide nanoparticles for antibacterial activity. Figure reproduced with permission of the publisher, 2016, Elsevier [43].

## 3. Blow-Spun Nanocomposites

### 3.1. Nanoparticles

One mechanism for functionalizing blow-spun fibers is the incorporation of nanoparticles. Titanium dioxide, copper, iron, silver, and zinc nanoparticles have been added to polymers and blow spun to fabricate coatings, films, scaffolds, and other constructs [43–45]. Gonzales-Benito et al. (2016) used SBS to prepare PVDF/TiO$_2$ nanocomposites (Figure 2) [43]. They further characterized their nanocomposite's surface properties and showed that PVDF/TiO$_2$ films enhanced antimicrobial activity against *S. mutans* adhesion. A PVDF/Ni, a fibrous material using the SBS technique with unusual magnetic properties, was produced by Diaz et al. (2018). The authors suggest that such a composite may have applications in different areas of the technological industry [44].

A composite of poly(lactic acid)/titanium dioxide (PLA/TiO$_2$) blow-spun fibers was produced by Costa et al. (2016) [45]. Various concentrations of TiO$_2$ were studied for their influence on fiber material properties. SEM and TEM analysis showed that the SBS method produced PLA/TiO$_2$ nanofibers with uniform morphology and without beads. Furthermore, the incorporation of TiO$_2$ nanoparticles could influence the PLA nanocomposite crystallinity. PLA photocatalytic degradation experiments revealed that PLA weight loss increased with TiO$_2$ content.

Hematite blow-spun fibers were produced using SBS modified with a microfluidic gas flow-focusing nozzle that does not require an external fiber pulling force [46]. Compared to other methods, this technique is inexpensive and user-friendly and permits precise fiber diameter control, high production rate (m/s-range), and direct fiber deposition without clogging due to stable, gas-focused jetting. In addition, iron nanoparticles were also incorporated into SBS fibers as a means for creating personal protective equipment [47]. The

resulting hematite nanocomposite fibers highlight this technology's exciting possibilities that can lead to the fabrication of multifunctional/stimuli-responsive fibers with thermal and electrical conductivity, magnetic properties, and enhanced mechanical stability.

Nanocomposites, comprised of silver nanoparticles (AgNPs) and polyaniline (PANi), were synthesized into plasma-pretreated cellulosic nanofibers fabricated by the solution-blowing spinning technique [48,49]. AgNPs and PANi enhanced the electrical conductivity of cellulosic nanofibers (CNFs). In addition, the produced CNFs demonstrated high UV protection and a potent antibacterial effect.

*3.2. Nanoclay*

Nanoclays have been studied as bulk fillers to enhance fiber material properties, drug delivery, and tissue engineering [50,51]. Nanoclays have also been explored for specialized applications such as battery storage. Organic montmorillonite and polyamide (PI) composite fibers were blow spun as a lithium-ion battery separator. Montmorillonite (MT) significantly enhanced the PI separator's mechanical strength (compared to unmodified PI separators) [52]. The MT-prepared hybrid separators showed superior electrolyte wettability and high ionic conductivity; no thermal shrinkage occurred at 180 °C for 30 min, a specific discharge capacity of approximately 117.2 mA h g$^{-1}$, and a good capacity. The study reveals a novel solution using a blow-spinning PI/MT hybrid separator for energy storage.

Halloysite nanotubes (HNTs) are extensively used as nanocontainers, nanocarriers, and nanoparticles that can modify material properties as needed (thermal resistance, mechanical properties) [53–56]. HNT nanotubes can be loaded with various materials for sustained and extended release and exhibit high levels of biocompatibility [54–56]. HNTs can modify the material properties of a polymer (durability, mechanical properties, thermal and wear resistance). HNTs have been used as nanocontainers [57,58] and nanocarriers [59]. HNT has negative and partially positive surface potential in HNT lumenal surface, leading to enhanced polymer solution conductivity [60]. The HNT nanotube can be loaded with various materials for sustained and extended-release and high levels of cytocompatibility [55–60]. HNTs have been used to develop blow-spinning applications (Figure 3). However, the agglomeration of nanoclay fillers is detrimental to the mechanical properties of polymer-based nanocomposites [60]. HNTs have also been incorporated into PVDF [61,62]. HNTs were used as an additive to PVDF and showed that HNTs aligned themselves along the fiber axis [63]. Additionally, the fabricated nanofibers were fine, smooth, and uniform, with an overall mean fiber diameter decreasing significantly after HNT addition [63].

Blow spinning enabled multifunctional applications of HNTs embedded in polymer scaffolds and airbrushed as fibers or films (herein called constructs). Film thickness and fiber orientation can be controlled. Single or multi-polymer constructs, as well as polymer composites such as mixed films and fibers or multilayered constructs with different polymer layers can be created. Adding a doped halloysite enables the development of a diverse range of functionalities. Boyer et al. (2015) [27] explored a range of potential polymers and their use with blow-spinning. All these studies have established the potential of nanofiber deposition onto any substrate with halloysite offering numerous opportunities for new and novel applications.

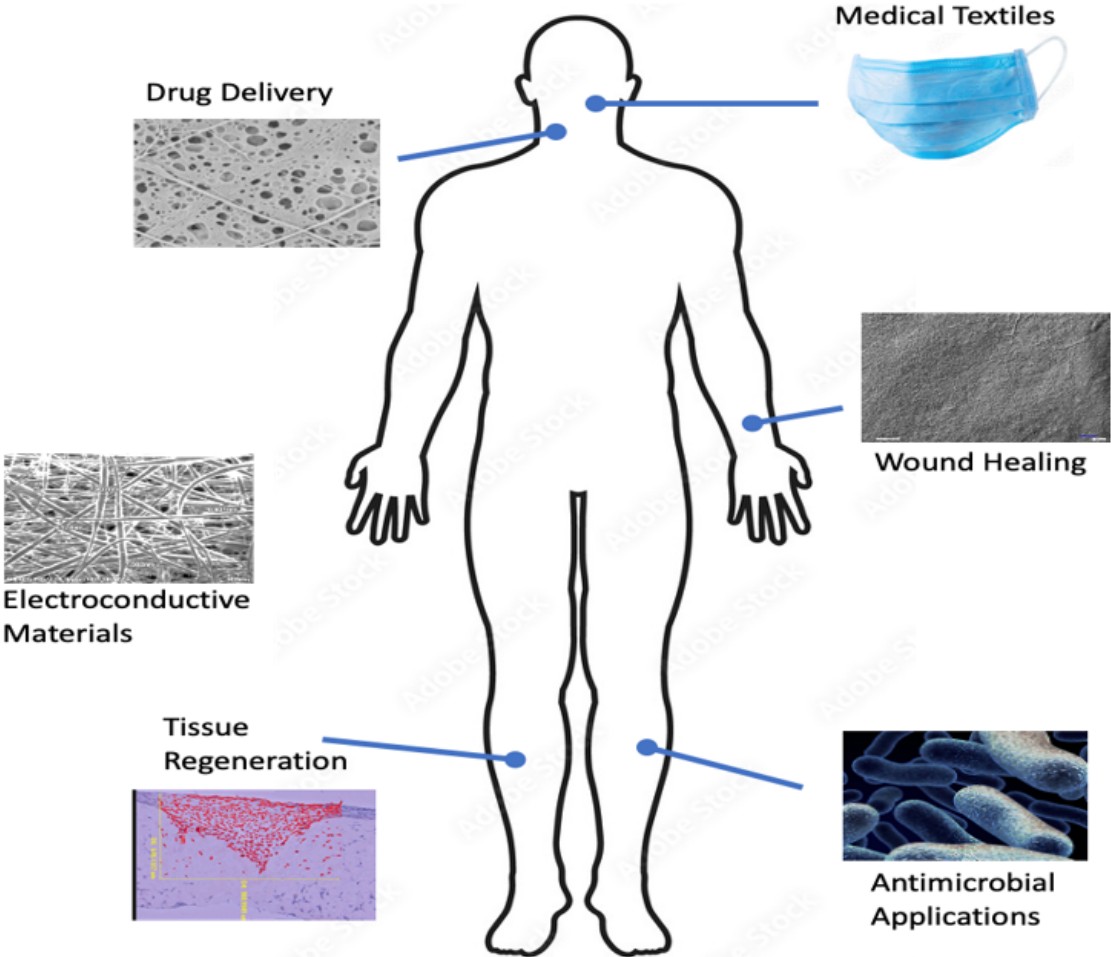

**Figure 3.** Applications of blow-spun fibers include medical textiles, drug delivery, wound healing, and tissue regeneration.

## 4. Blow-Spun Applications

### 4.1. Antimicrobial

Eliminating the devastating societal impact from viral pandemics and bacterial and fungal infections is a critical need [64]. Solution blow spinning has an important part to play in advancing antimicrobial biomedical applications and materials. PCL, PLA, cellulose nanofibers produce through SBS have been employed in various antimicrobial applications [65,66]. These collaborative endeavors will lead to greater treatment modalities for the treatment and prevent the spread of infectious disease [64–66]. Figure 4 illustrates recent use of mHNTs)/PLA blow-spun fibers used as part of a filter system in an N95 mask [67].

### 4.2. Drug Delivery

Drug delivery systems as a new strategy for the treatment of various kinds of diseases have been used during the last decades because this strategy has different advantages such as releasing therapeutic treatment at the specific target, and the ability to control the rate of release, improved efficiency, lower side effects, and safer administration [68]. One of the best scaffolds for delivering a different drug is nanostructure scaffolds containing porous and nonporous fibers [69]. For this purpose, various researchers have been designing and fabricating functional scaffolds with the blow spinning method for delivering drugs. As an illustration, Bonan et al. [70] used solution blow-spun poly(lactic acid)/polyvinylpyrrolidone micro-and nanofibers to fabricate a carrier for delivering Co-

paiba oil. The main reason for using polyvinylpyrrolidone was to enhance fiber diameter and reduce surface contact angle.

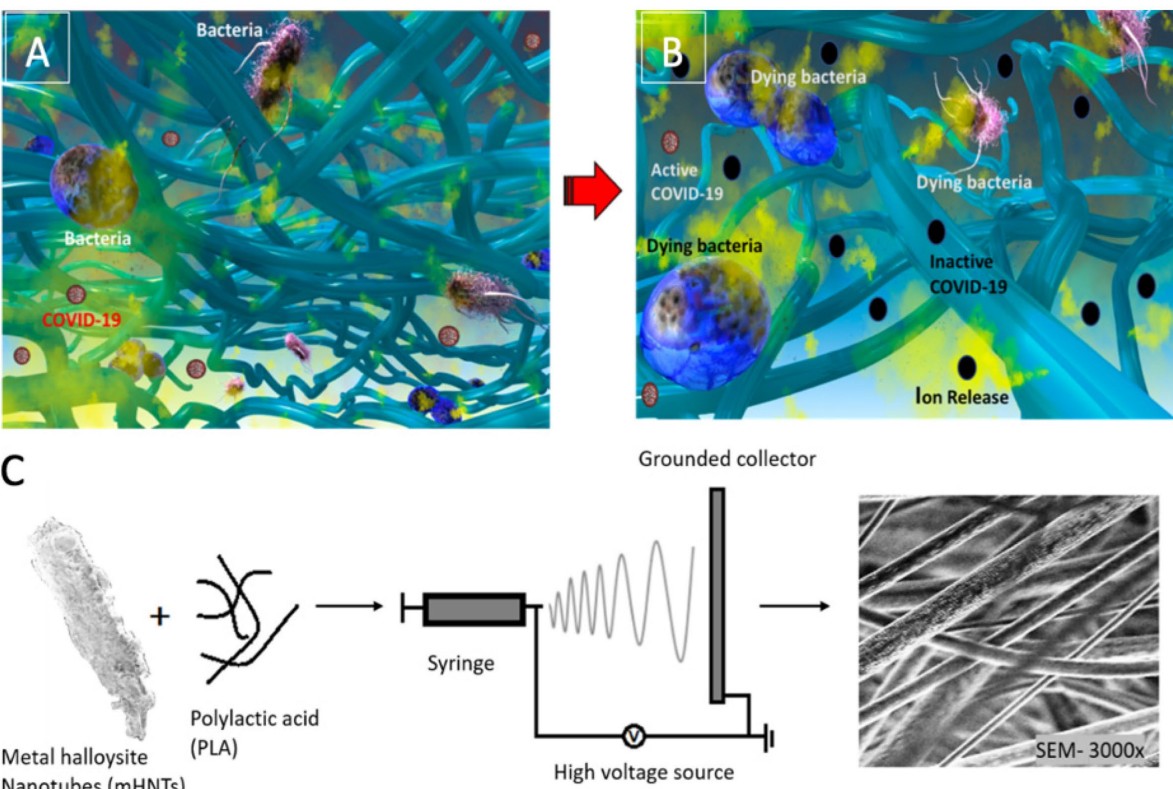

**Figure 4.** (**A,B**) Graphic depicting metal-coated HNTs (mHNTs) embedded in blow-spun fibers used to inhibit bacterial growth and inactivate viruses. (**C**) The mHNTs and PLA electrospun fibers were used as part of a filter system to be used in N95 mask [67].

Additionally, polyvinylpyrrolidone fibers showed a higher release rate of Copaiba oil, and nanofiber mats containing the drug exhibited antimicrobial properties. Finally, it should be mentioned that by increasing the amount of polyvinylpyrrolidone, the antimicrobial property of the scaffold was improved significantly. As a result, researchers could change the release rate of drug, antimicrobial, and morphological properties by changing the nanofiber composite. In another study, Oliveira et al. [71] fabricated nanofibrous mats based on PLA to control progesterone delivery. In this study, researchers have used various methods such as SEM, FTIR, differential scanning calorimetry (DSC), and characterizing the spun membranes. As a result, they could fabricate fibers with diameters from 280 to 440 nm to deliver various progesterone concentrations.

Furthermore, it was demonstrated that solution blow spinning could be used only to control the delivery of progesterone but also to encapsulate an active agent into polymer fiber. Recently, to control delivering amphotericin B, Gonçalves, et al. [72] fabricated core-shell nanofibers based on poly (L-lactic acid) and poly (ethylene glycol) with a solution blow spinning method. As a result, they could design outstanding polymeric nanocarriers with smooth surfaces and uniform morphology. Consequently, the solution blow spinning method could be an important and valuable choice for fabricating different drug delivery systems to treat various diseases. Figure 5 illustrates a concept for a multifunctional bandage that could be used for cutaneous or mucosal drug delivery, injuries, or as an anastomoseal after abdominal organ surgery.

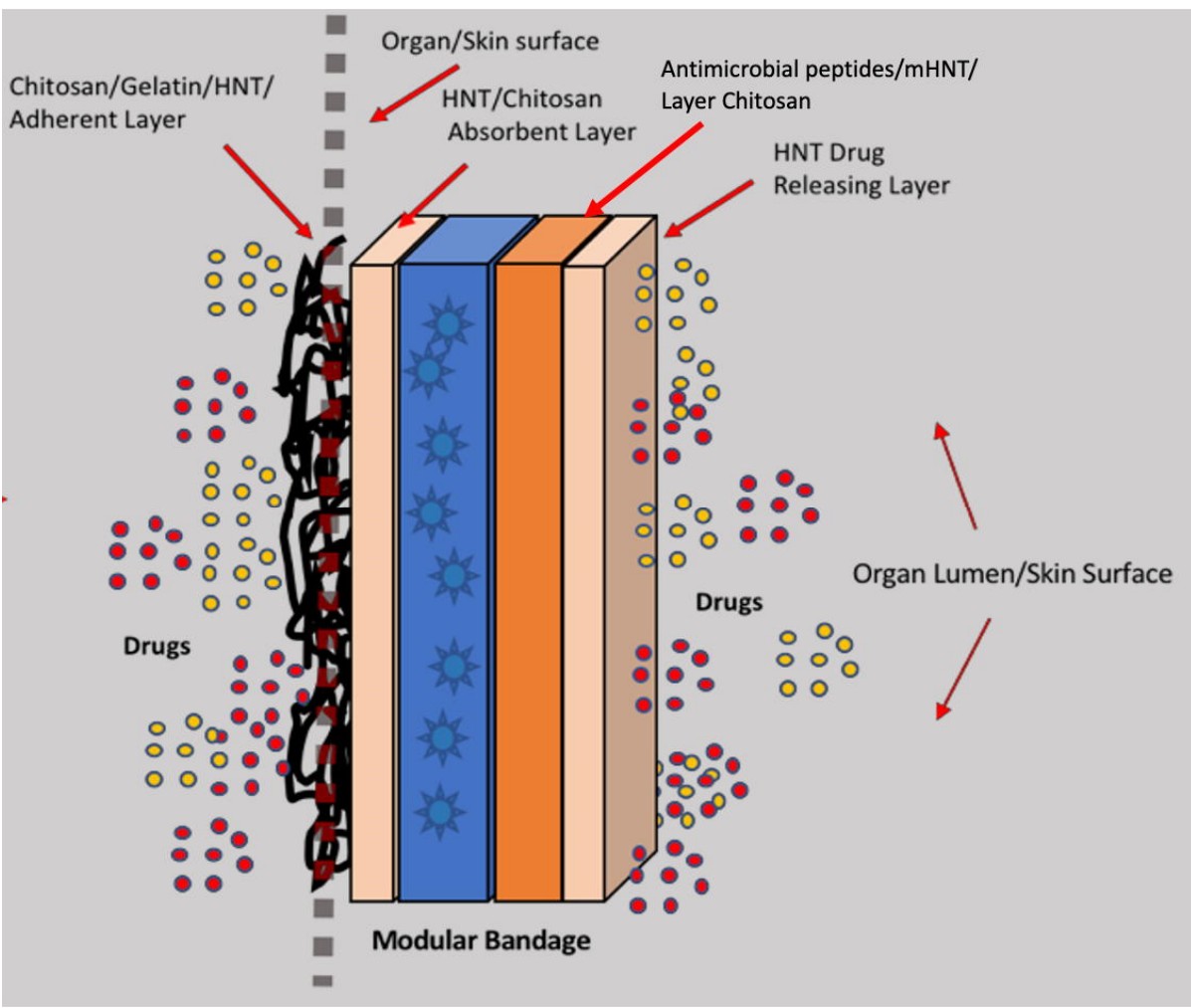

**Figure 5.** Conceptual idea for a multifunctional bandage. From the communicating author's collection.

### 4.3. Textiles Used in Medicine

The term "medical textiles" describes quality-assured products made of fibers obtained from natural or synthetic polymers and used for medical purposes, including but not limited to surgeries, enhanced hygiene, and clinical treatments [73]. Blow-spun polymer fibers can be further functionalized by incorporating additives such as nanoparticles [74], hydrogels [75], and bioactive compounds [76] to enhance their inherent properties. Medical textiles are widely used in the healthcare and well-being sectors. Current applications of medical textiles encompass wound care [77], biomedical [78], as well as conductive and electronic applications, combining textile science and electronics [79]. Medical textiles are primarily woven, knitted, or braided fabrics and have the advantages over non-woven forms in wound healing and tissue regeneration [77,78].

Blow-spun medical textiles are currently in increasing demand because solution blow spinning and its modifications enable reproducible fabrication of medical textiles with precise control over microstructure [80,81]. Blow-spun medical textiles are relatively cheap and allow for the production of multicomponent fibers owing to the non-sophisticated method of fabrication [82,83]. Among the commonly used materials for medical textiles are ultra-high molecular weight polyethylene, polypropylene, alginate, and nylon [84].

Trauma

The victims of traumatic events incur injuries in hard-to-bandage areas that are often prone to profuse bleeding due to a high degree of vascularization, for example, head

injuries and wounds that penetrate internal body cavities [84]. Ineffectual treatment options for injuries can often result in a number of post-injury complications including (1) lack of hemostasis in the critically injured leading to severe blood loss; (2) extended convalescence periods resulting in increased medical costs for families; and (3) additional blood transfusions, which increase the stress on the already taxed blood supply [85].

Bacterial infection can further complicate healing and infection, which poses a significant threat to public health. Therefore, a bandage providing sustained release of antibiotics/antiseptics directly into the wound site is highly desired, and nanoparticles have been proposed as carriers for these agents. The advantage of using a blow-spun bioactive bandage is that it can prevent/remediate an infection and minimize the building up of scar tissue and produce a more cosmetically appearing healed wound [86]. Solution blow spinning research to create a conductive nano-enhanced wound dressing using cellulose, PCL, or PLA fibers infused with copper or silver nanoparticles is under intense investigation. These conductive materials could be used for inhibiting/eliminating bacteria as well as assisting with neotissue formation [87,88].

### 4.4. Tissue Regeneration

Tissue regeneration is a promising approach to fulfill the chronic shortage of human donors for tissue and organ transplantations. A vital element of this approach is the scaffold, which supports the seeded cells and the desired mechanical properties for biocompatible tissue and has tunable mechanical properties suitable for each specific application. Current research efforts are directed at synthesizing new biomaterials with improved chemical, biological and mechanical properties. Wound healing and bone tissue engineering will illustrate this effort.

### 4.4.1. Skin Tissue Regeneration

SBS has been used to prepare different kinds of biocompatible fibrous materials for skin tissue regeneration, and with this technique, scientists worldwide could affect the speed of wound recovery. Liu et al. used chitosan/polyvinyl alcohol (CS/PVA) hydrogel nanofiber mats as a unique scaffold for wound healing applications [89]. In this study, researchers utilized ethylene glycol diglycidyl ether (EDGE) as a cross-linker to improve the properties of the hydrogel nanofiber mats. In this research, to analyze the effect of cross-linker on different properties of nanofiber mats, nanofibers with non-exact cross-linker content have been prepared and analyzed.

For example, Lorente et al. [41] recently used this method to fabricate a scaffold poly-ε-caprolactone (PCL). This research modified the biocompatible fibrous with collagen to improve different properties. For instance, adding a small amount of collagen, topography, morphology, wettability, cell adhesion, and proliferation of the scaffold have changed meaningfully.

In another study, the researcher fabricated adhesive wound dressing with antimicrobial properties for tissue regeneration and remodeling. For this purpose, they used chitosan/polyethylene oxide nanofibers as biodegradable polymers and antimicrobial silver to improve antibacterial properties of their wound dressing [90]. Additionally, scaffolds without antimicrobial silver showed low cytotoxicity, but when antimicrobial silver was loaded into a dressing, antimicrobial silver showed a controllable releasing rate from 7 to 14 days, and the antibacterial properties of the blow-spun scaffold improved. The nanofiber mats fabricated with the solution blowing process exhibited smooth surfaces and good antibacterial properties against *Escherichia coli*.

The possibility of direct deposition introduces a host of new applications and advantages over preformed nanofiber mats/meshes and scaffolds [91]. Centrifugal spinning fabrication of conformal nanofiber mats/meshes allows for precise and site-specific construction [92]. Like solution blow spinning, centrifugal spinning is a simple method for producing fibrous scaffolds with large and interconnected pores. Functionalization of the fibers enhances their antimicrobial and regenerative properties. These capabilities could

be exceedingly helpful for reconstructing tissue defects such as hernias, which frequently use preformed polymer mats with a high incidence of recurrent herniation and bowel obstruction. The approach also holds great promise as a surgical sealant in place of or in addition to sutures in vascular, intestinal, or airway anastomosis applications. However, these procedures can be technically challenging and produce morbidity when leakage complications occur.

SBS could also be valuable in areas requiring the use of a hemostatic material or sealant, especially when large areas are exposed and conventional suturing may not be possible, as is the case with liver and lung resections [93]. A spray-on enables the application of any wound geometry (custom fit dressing) to replace sutures or bandages conceivably [93]. Nanofiber bandages have been under intensive study, but traditional methods of applying them using "electrospinning" would not be feasible. The airbrush method is clean since the solvent required to produce the polymer nanofiber sprayable evaporates before hitting the target and decays to nothing in 42 to 160 days, depending on conformation [94].

4.4.2. Bone Tissue Regeneration

Bone defects resulting from trauma, disease, surgery, or congenital abnormalities are a worldwide health problem. In 2020, bone was the second most common transplanted tissue, with an estimated 2.7 million grafting procedures annually [95]. Bone grafts (BGs) treat bone fractures, regenerate bone lost due to trauma or infection, or repair and rebuild diseased bones in a human body that cannot heal the bone by itself [96]. An alternative is bioactive scaffolding for bone tissue regeneration. The critical properties needed for successful bone tissue regeneration include an interconnected porosity with pore sizes that support the diffusion of nutrients and exchange of waste products and migration of bone cells, bioactivity that encourages vascularization and bone formation, and controlled biodegradability that ensures resorption and remodeling [97]. Finally, the scaffold should possess sufficient dimensional stability enabling it to adapt to the unique features of the defect. Furthermore, the ideal scaffold should contain a combination of biomaterials with varying osteoconductive features and be able to support new and healthy three-dimensional bone formation. An implantable scaffold should significantly improve osteogenic potential so that recovery time is reduced, and healing is promoted for critical-sized bone defects [98]. Scaffolds with additional functionalities such as antimicrobial or chemotherapeutic properties are also needed.

Composite poly-L-lactide acid (PLLA)-based scaffolds with hydroxyapatite content of up to 75 wt.% hydroxyapatite was fabricated via solution blow spinning [99]. The influence of hydroxyapatite concentration on the structure, wettability, mechanical properties, and chemical and phase composition of the fiber composite was examined. The authors showed that increasing hydroxyapatite content increased the average fiber diameter, uniaxial strength and elongation were reduced, and surface wettability did not change. Implantation of the bone in a rat calvarial defect model in the parietal bone of a rat skull was studied over 90 days. Scaffolds with 25 wt.% hydroxyapatite content significantly enhance osteogenesis and may be highly efficient for replacing bone defects [99]. Core-sheath fibrous poly-ε-caprolactone (PCL) core and polyvinyl alcohol (PVA) scaffolds with hydroxyapatite embedded were fabricated by Mahalingham et al., 2021 [100]. Results showed that the fibers were not cytotoxic, while a significant increase in cell proliferation was obtained for PCL–PVA/5%/HA fibers [100].

PLGA (poly(lactic-co-glycolic acid) is an FDA-approved copolymer which has been widely investigated in developing bone implants and scaffolds due to its biocompatibility and biodegradability. Boyer et al. (2015) demonstrated that halloysite nanotubes (HNTs), a recognized nanocontainer and nanocarrier, could be embedded in blow-spun fibers as three-dimensional scaffolds for bone tissue generation (Figure 6) [101]. Pre-osteoblasts proliferate and produce a matrix rich in glycosaminoglycans and collagen.

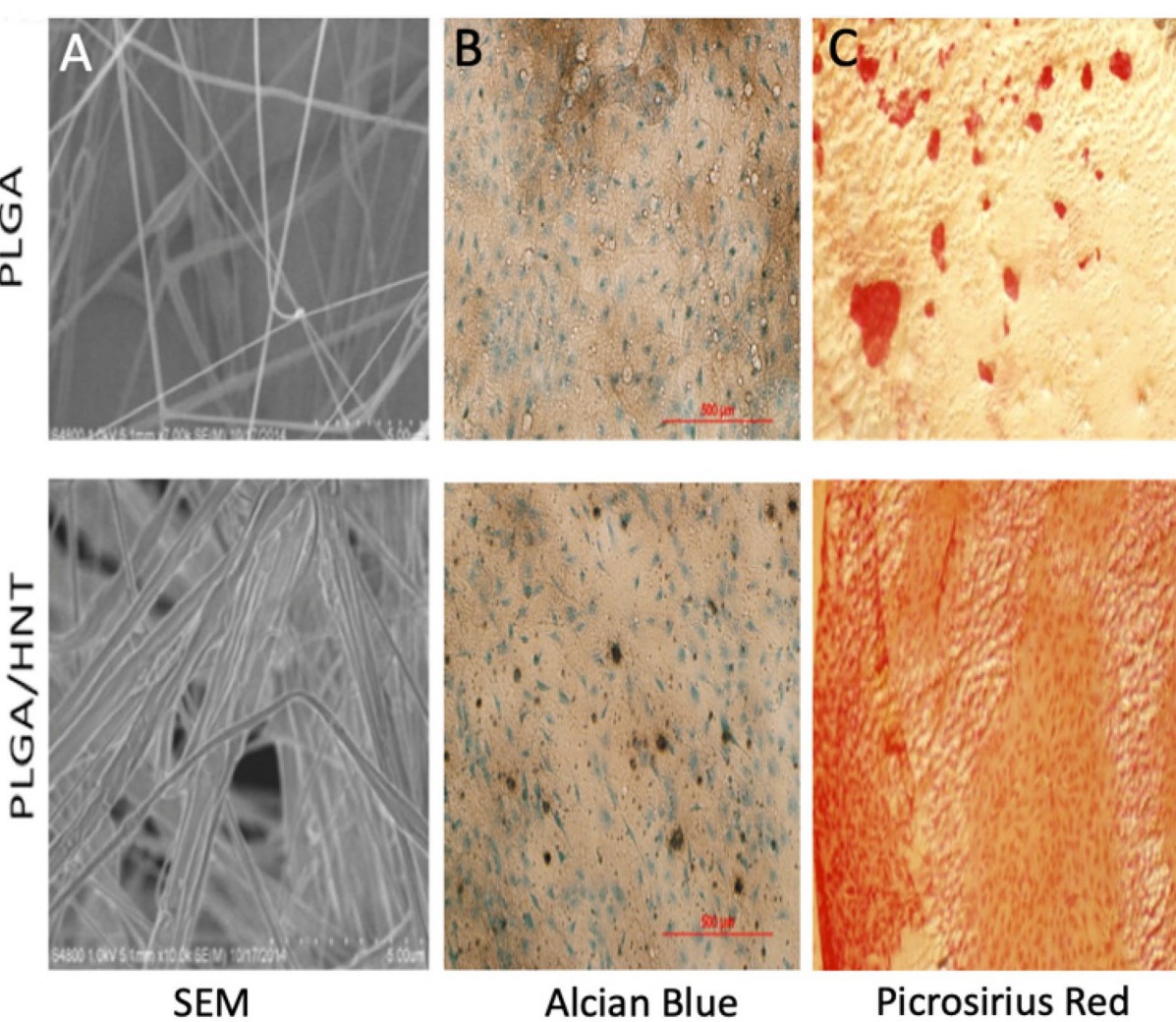

**Figure 6.** (**A**) SEM micrographs of blow-spun fibers with and without PLGA. Pre-osteoblasts were cultured on blow-spun fibers for 21 days. (**B**) Blow-spun fibers (+/− HNTs) stained with Alcian blue on day 3. (**C**) Blow-spun fibers (+/− HNTs) stained with Picrosirius red on day 3.

## 5. Commercial Applications

Continuous advances in the field of cell and tissue engineering offer scientists hope for the future developments of implantable tissue. Bone, cartilage and skin tissues have already been commercialized. Electrospinning and electrospraying are methods that can be scaled up for the various industries. With electrospinning, the process for fiber production into bandages, gauze and medical textiles are transformable into large-scale, high-throughput manufacturing processes. SBS can also be scaled for different applications, from small-scale company applications to a portable (transportable) carry-size model for use on battlefield or with emergency medical teams in a small gun-like system controlled by a microprocessor and equipped with $CO_2$ canisters for fiber delivery.

Biomedical textiles as fibrous structures are designed for specific applications and modified with additives that possess antimicrobial, fire-resistant, hemostatic, and tissue regenerative properties. These include implantable applications made of biodegradable fibers that can be used in specific environments; they are inherently biocompatible, biostable, and biodegradable. The non-implantable application includes appliques, bandages, drapes, and gauzes used in a range of surgical operations. The biomedical textiles market size was USD 11 billion in 2017 and is predicted to reach USD 14.50 billion by 2022, at a CAGR of 5.7% over the period [102]. The increased growth in market demand has been attributed to

COVID-19 and the need for PPE, extracorporeal devices, an aging world population, as well as the increased number of surgeries performed each year [103].

The global nanofiber market size was estimated to be USD 785 million in 2021 [104]. By 2030, it is expected to grow to USD 3.35 billion, registering a CAGR increase of 18% between 2022 and 2030 [105]. North America holds the largest share, approximately 37% of the total nanofibers market, followed by Europe and then Asia. The demand is being driven by a global-wide need for medical-grade face protection, air and microfiltration media, high-efficiency fuel filtration, sophisticated filtration, and biomedical research media [106].

## 6. Limitations of Blow Spinning

Technical developments and new SBS methods will continue to advance the application of this technology in all areas. Ultimately, the further development of blow-spun nanofibrous scaffolds must address several issues, including solvent/crosslinker toxicity used, efficient drug delivery, tailored stiffness and fiber orientation, cell adhesion and migration, directed cell differentiation, and desired biodegradability.

Solvents in SBS present significant challenges in terms of high flammability, toxicity, and disposal costs; they are also dangerous to human health and can harm the environment. The mass production of fibers using various solvents will require specialized technical equipment for solvent recovery to limit human exposure. The toxicity and cost of most SBS solvents will necessitate solvent recycling on environmental and economic grounds. However, these processes can be both expensive and complex. Furthermore, the most commonly used solvents are restricted by the European Union regulations on chemical control (REACH) [107]. Therefore, many researchers are seeking alternative and "green" solvents such as dimethyl carbonate [108].

Controlled or sustained drug delivery from blow-spun scaffolds poses a significant challenge. The use of nanocontainers, such as halloysite, can provide sustained release [54–56]. The reproducibility and alignment of micro- and nanofibers can be produced with the proper SBS configuration, nozzle type, flow rate, or channeled airflow [109]. Therefore, every basic SBS setup must optimize fiber size and alignment, strength, porosity, and support cell proliferation, difference and histogenesis. It is anticipated that future commercial developments to the basic setup will result in better quality fiber mats and optimization of the process. A programmable delivery system for nanofiber-based regenerative scaffolds is a critical need and can resolving many of the issues raised above will open up fascinating new possibilities for tissue regeneration. Adaptation to large-scale industrial production using SBS remains a significant challenge. However, it remains one of the most industrially viable spinning systems for producing submicron/nanofibers for various applications.

## 7. Future Directions

The possibility of a drug-doped direct deposition fiber system introduces many new applications and advantages over performed nanofiber mats/meshes and scaffolds. On-demand fabrication of conformal nanofiber mats/meshes allows for precise and site-specific construction. In addition, drug selection can be customized regarding the type and drug load.

This could be exceedingly useful for reconstruction of tissue defects such as hernias, the treatment of which frequently requires preformed polymer mats with a high incidence of recurrent herniation and bowel obstruction [110]. These procedures can be technically difficult and, when complications of leakage occur, and have high morbidity rates [111,112]. There is potential to use blow-spun mats as a surgical sealant in place of or in addition to sutures in applications such as vascular, intestinal, or airway anastomosis. Solution blow spinning could also be useful in bowel surgery and as a hemostatic material or sealant, especially when large areas are exposed and conventional suturing may not be possible, as is the case with liver and lung resections [112,113].

**Author Contributions:** D.K.M. conceived of a mini-review paper; M.M., M.L., A.-R.M. and D.K.M. contributed to the writing. Figure design was completed by D.K.M. All authors contributed to editing the revise manuscript. All authors have read and agreed to the published version of the manuscript.

**Funding:** This research was funded by the NASA EPSCoR Project Office under grant award (21-EPSCoR-R3-0009) to D.K.M.

**Data Availability Statement:** Data cited in the paper is available from the references cited.

**Acknowledgments:** The authors acknowledge the blow-spinning research of students in Mills lab.

**Conflicts of Interest:** The authors declare no conflict of interest.

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
