# Peer review of "Biomedical Applications of Blow-Spun Coatings, Mats, and Scaffolds—A Mini-Review"

_jcs, doi:10.3390/jcs7020086_

Round 1

Reviewer 1 Report

Here are the suggestions before it could be accepted.

1. The literature should be updated, more literature should be in recent three years.

2. In the introduction, the disadvantages of the references should be summarized clearly to emphasize the importance of this work.

3. “Type of the Paper (Article, Review, Communication, etc.) ” The type of this paper should be selected.

4. The resolution of figure 5 should be improved.

5. I suggest some quantifiable results should be added in the conclusion.

6. All the pictures cited from references should mark the reference number and copyright permission.

Reviewer 2 Report

The review paper entitled "Biomedical applications of blow spun coatings, mats and scaffolds” has been reviewed. The blow spinning development, nanocomposite and applications have been discussed. However, major revisions are needed to be satisfactory in order to be published in J. Composite Sci. Detailed comments are as follows:

1.       The current development, the elevated attention of blow spinning and the promising future in bio could be highlighted in the abstract.

2.       The introduction section is general and brief. The state-of-the-art methods, the current challenges and the focus of this paper are missing, which needs to be added.

3.       In this review paper, the comparison of blowing spinning techniques and other existing ones is insufficient. The advantages/disadvantages need to be thoroughly compared.

4.       In the section on blow-spun nanocomposites, different types of materials should be blended organically based on their properties, features and applications rather than being simply surveyed.

5.       The fundamental technical challenge of blow spinning for further development needs to be specified, in order to be published in J. Composite Sci.

6.       Tissue regeneration Sec.4.4 overlaps with wound healing Sec.4.4 and needs restructured.

7.       More discussions on Coatings, Mats, and Scaffolds would be great.

Round 2

Reviewer 1 Report

It can be accepted in this form.

Reviewer 2 Report

The concerns have been well addressed by enriching the contexts and targeting a mini-review. The format of Refs 82-88 needs to be fixed.